# The role of social determinants in COVID-19 hospitalization disparities by migration status in Stockholm, Sweden. A population-based cohort study

Yan Ma ⬤[1,2] ✉, Anders Ledberg[2,3], Siddartha Aradhya ⬤[4] & Sol P. Juárez[1,2,5] ✉

## Abstract

**Background** Immigrants in Sweden, particularly those from low- and middle-income countries, had higher risks of COVID-19 mortality and morbidity compared to the Swedish-born. However, prior studies have not quantified the contribution of the differential distribution of health and social determinants to the increased risks. **Methods** We used total population registers from Sweden to investigate disparities in COVID-19 hospitalization between five groups of immigrants and Swedish-born, using a cohort 577911 working-age adults (18–65 years) living in Stockholm during the first two waves of the COVID-19 pandemic. Applying a decomposition analysis, we quantified the relative contribution of age, sex, income, education, occupation type, residential area, and pre-existing medical conditions to these disparities. **Results** Our study shows that immigrants have higher risks of hospitalization compared to Swedish-born, and that the investigated factors accounted for these disparities to varying degrees across immigrant groups. For the most affected immigrant groups (from Africa and Middle East), the examined factors together account for only a minor part of the disparities (21% and 18% for Wave 1; 16% and 11% for Wave 2), with occupation type and residential area contributing substantially. **Conclusions** Common observable social determinants of health account for a moderate share of the overall disparities in COVID-19 hospitalizations between Swedish-born individuals and immigrant from the most affected regions of origin.

## Plain language summary

The COVID-19 pandemic has disproportionately affected foreign-born individuals, who have experienced higher COVID-19–related mortality and morbidity compared to the majority populations in the receiving countries. Using mathematical analysis, we estimated how much of the difference in COVID-19 hospitalization risk between immigrants and the majority population in Sweden can be explained by various factors associated with adverse COVID-19 outcomes, including age, sex, health, and social factors. Our results show that the studied factors account for the immigrant–native differences in hospitalization to varying degrees across immigrants' regions of birth. These factors explained the least proportion of this difference among the groups most affected by COVID-19. This highlights the need to continue identifying factors underlying immigrants' higher risks of adverse COVID-19 outcomes.

Immigrants (i.e., foreign-born) in many countries were disproportionately affected by the COVID-19 pandemic[1–4]. In Sweden, immigrants, particularly from low- and middle-income countries, such as Iran, Iraq, and Somalia, had significantly higher mortality rates and more severe complications during the pandemic compared to the Swedish-born[5–8]. The greater impact of the pandemic on immigrants, both in Sweden and other countries, was in stark contrast to their often lower mortality compared to native populations prior to the pandemic[9–12].

Differences in COVID-19 outcomes between immigrants and the Swedish-born have been reported for both working- and retirement-age populations, with higher risk ratios among the working-age population[8]. These differences remained after adjusting for health and socio-demographic variables—factors likely reflecting a mixture of differential exposure and susceptibility[5,8,10]. However, these studies did not quantify the relative contribution of each factor, nor did they examine in detail the possibility that the same set of health and social factors may impact the risk

[1]Centre for Health Equity Studies (CHESS), Stockholm University/Karolinska Institutet, Stockholm, Sweden. [2]Department of Public Health Sciences, Stockholm University, Stockholm, Sweden. [3]Centre for Social Research on Alcohol and Drugs (SoRAD), Stockholm University, Stockholm, Sweden. [4]Stockholm Demography Unit, Department of Sociology, Stockholm University, Stockholm, Sweden. [5]Department of Global Public Health, Karolinska Institutet, Stockholm, Sweden. ✉e-mail: Yan.ma@su.se; Sol.juarez@su.se

differently for different immigrant groups. In other words, previous studies have not completely acknowledged the heterogeneity of the immigrant groups in Sweden, as they implicitly assume that all socio-demographic variables have the same "effects" across all origins.

The aim of this study is to apply a decomposition analysis to quantify the contributions of several factors to COVID-19 hospitalization risk differences between immigrants and the Swedish-born population residing in Stockholm Municipality during the first two waves of the pandemic (March 2020 to January 2021). We focus on the working-age group since the relative risk is higher in this group, and moreover, occupation can be used as an indicator for COVID-19 exposure in this group. Overall, this study shows that immigrants had higher risks of COVID-19–related hospitalization compared to the Swedish-born, with large unexplained disparities remaining for groups from Africa and the Middle East. These findings highlight the need for efforts to identify the determinants underlying the differences in hospitalizations risks between these groups and the Swedish-born.

## Methods
### Study population and data
This study focuses on the first two waves of the COVID-19 pandemic, prior to the mass vaccination period. Here we consider the first wave (Wave 1) from March 1, 2020 to August 31, 2020, and the second wave (Wave 2) from September 1, 2020 to January 31, 2021.

The study population comprised all working-age individuals (18–65 years old in 2020) who were alive and had a registered residential address in Stockholm Municipality as of 2019, totaling 640,207 individuals. We excluded 15,650 individuals with missing records of country of birth. To ensure that pre-existing medical conditions were measured over the same period of time for all individuals, we included only those who had continuously resided in Sweden for at least five years before the onset of the pandemic (2015–2019). This criterion led to the exclusion of an additional 46,065 individuals, among whom 64 and 56 were hospitalized due to COVID-19 in Wave 1 and Wave 2, respectively—corresponding to lower hospitalization rates compared to the remaining population. Individuals who died before the start of the pandemic were excluded (150). We also excluded individuals who died from COVID-19 without prior hospital admission or from other causes during the respective wave (431 in Wave 1, and further 310 in Wave 2 after excluding 64 people died in wave 1, out of which 20 and 3 died from COVID-19 without prior hospital admission). After applying these exclusion criteria, the final study population comprised 577,911 individuals for Wave 1 and 577,537 for Wave 2.

Data were obtained from multiple national registers linked via unique pseudo-anonymized identification numbers. Data on mortality, including dates and causes, were retrieved from the Cause of Death Register (CDR). Data on inpatient care (hospitalizations), with dates and diagnoses, were obtained from the National Patient Registry (NPR). Data on individual sociodemographic background, including year of birth, sex, and residential areas, and country/region of birth was obtained from the Total Population Register from 2019. Data on occupation, disposable income, education attainment were retrieved from the Longitudinal Integrated Database for Health Insurance and Labor Market Studies (LISA).

### Outcome
COVID-19 hospitalization (hospitalized vs. not hospitalized) was identified through the main diagnosis recorded in the National Patient Register (NPR) with the corresponding ICD codes: U07.1, and U07.2.

### Migration status
Immigration status was defined by the country of birth, and was grouped into seven geographic countries/regions: *Sweden, Other Nordic Countries* (Finland, Denmark, Norway, and Iceland excluding Sweden), *Other European Countries* (including Russia), *Middle East* (including Turkey), *Africa*, and *Rest of the World*. Hereafter, the names of regional groups (e.g., "*Other Nordic Countries*") refer to immigrants from the countries included in each

respective group, unless stated otherwise. Countries/regions within each regional group can be found in Supplementary Table S1.

### Covariates
Age was calculated as the difference between 2020 and the individual's birth year. Pre-existing medical conditions associated with severe COVID-19 outcomes[13] were assessed using information from the National Patient Registry. In particular we constructed a binary variable indicating whether an individual had at least one of the following main diagnosis recorded in within five years prior to the pandemic (January 1, 2015 – December 31, 2019): chronic kidney disease (ICD-10 code: N18), diabetes (E08–E13), cardiovascular diseases (CVD: E78, G45–G46, I10–I13, I20–I26, I63–I66, I80–I82), neurological diseases (G20, G21, G30, G35–G37), chronic respiratory diseases (J40–J47), tuberculosis (A15–A19), HIV (B20–B24), chronic liver disease (K70–K77), cancer (C00–C96) and sickle cell disease or thalassemia (D56–D59). Occupation type from 2019 was classified into three categories, "essential occupations", "non-essential occupation", "unemployed or unregistered". Essential occupations were defined as in Billingsley et al.[14], and included care workers, police officers and security guards, service sector personnel, delivery workers, taxi- and bus drivers, teachers, meat packers, and cleaners (based on Swedish standard SSYK 2012, codes can be found in Supplementary Table S2). Individuals with a registered occupation that was not classified as essential were categorized as "non-essential workers." The "unemployed or unregistered" category included individuals who did not have an occupation registered in the system, including not only unemployed individuals but also those working in unreported jobs or informal employees, such as part-time employees in restaurants. Disposable income from 2019 was divided into tertiles: low-, medium-, and high-income levels. Education attainment (according to Swedish Education Classification SUN2020) was classified into primary education (9 or fewer years of education), upper secondary education (10–12 years of education), post-secondary education ( > 12 years of education) and missing (individuals with missing educational attainment). As an indicator of area of residence, we used "District", a nationwide geographical division corresponding to the parish division as on December 31, 1999. The municipality of Stockholm has 28 districts with areas ranging from 0.56 km$^2$ to 17.63 km$^2$, and populations from 4584 to 59154 inhabitants. Given that the outcome we study is relatively rare, a tessellation with smaller geographical regions was deemed not meaningful. To determine which district someone resided in, we used information on which demographic statistical areas, DeSO (demografiska statistikområden in Swedish) someone resided in.

### Statistical analyses
Hospitalization risk was taken as the number of hospitalized people divided by corresponding population in the beginning of each wave. To examine the association between country/region of birth and COVID-19 hospitalization, we estimated both unadjusted and adjusted odds ratios (ORs) using logistic regression. The regression models assessed whether being an immigrant from a specific country or region was associated with risk of hospitalization, both without and with adjustments for all covariates of interest. Given that hospitalization risks are small (in the order of less than 1%), we used odds and risks synonymously in the remaining.

To quantify the individual contribution of the factors to the hospitalization risk difference, we applied a decomposition analysis. This method is an extension of the Oaxaca-Blinder decomposition[15,16] to nonlinear models[17], including logit and probit models, making it suitable for the binary outcomes studied herein. The aim of this decomposition is to explain the hospitalization risk difference by decomposing it into two components: *the composition effect*, accounting for differential distribution of the measured factors, and; *the coefficient effect*, accounting for the remaining risk difference. The interest lies in the composition effect which quantifies how much of the hospitalization risk differences can be attributed to differences in how the measured factors are distributed between immigrant groups and the Swedish-born. For example, age is a major risk factor for complications of

COVID infections, and the composition effect quantifies the extent to which differences in age structures between two groups lead to differences in hospitalization risks. Intuitively, a large coefficient effect means either that the model is incomplete or that different factors have different effects in different groups. However, the contribution of each factor to coefficient effect cannot be quantified due to the nonlinear link function in the logistic regression[17]. To still get some insight into this issue, we fitted logistic regression models to data for each group separately, of which the results are shown in Supplementary file and briefly discussed in the next section. To ensure a robust estimate of coefficients to calculate composition effect, we used pooled coefficients, i.e., coefficients from logistic regressions including all study population with dummy variables for regional groups. By doing so, we assume the pooled coefficients are the reference coefficients or a non-discriminatory condition where all population following the same effects on included factors. Details of this method are described in the Supplementary file.

All factors added into the analysis were quantified or categorized as described previously, except for a higher-order term of age (age[2]) that was also added into logistic regression and later decomposition analysis to account for non-linearity in the relationship between age and hospitalization.

Data management was performed using Python 3.12.1. Logistic regression and Fairlie decomposition analysis were conducted in STATA MP 15.1 using the Fairlie software package[18].

### Ethics and inclusion statement

The study has received approval from the Regional Ethical Review Board of Stockholm, that also waived the need for informed consent due to the anonymized nature of the data material (decision no. 2022-00428-01). The data were processed in accordance with the General Data Protection Regulation and the research was performed in accordance with the relevant guidelines and regulations (Declaration of Helsinki).

## Results

### COVID-19 hospitalization risk and associated factors by country/region of birth and epidemic wave

Table 1 presents the COVID-19 hospitalization risk and the distribution of covariates across groups defined by country/region of birth. Figure 1 shows the hospitalization risk across districts. Hospitalization risk was higher among all immigrant groups compared to the Swedish-born, in both waves. In Wave 1, the Swedish-born had a hospitalization risk of 0.14%. Among immigrants, the highest risk was observed in *Middle East* (0.79%), followed by *Africa* (0.68%), *Rest of the World* (0.49%), *Other Nordic Countries* (0.32%), and *Other European Countries* (0.21%). In Wave 2, hospitalization risk decreased for all groups compared to Wave 1, but *Middle East* continued to have the highest risk (0.48%). These large differences in hospitalization are much greater than for infectious and parasitic diseases or for respiratory system diagnoses (excluding COVID-19) before and during the pandemic (Supplementary Fig. S1).

Statistics in Table 1 show that the measured factors are differentially distributed across immigrant groups compared to the Swedish-born. Notably, some immigrant groups have a much higher percentage of essential workers than the Swedish-born population, and also show higher residential clustering. For example, 44.9% of immigrants categorized in the group *Africa* have occupation belonging to the category of essential workers, compared to 17.1% among the Swedish-born, and almost half of them resided in three out of 28 districts (Spånga, Kista, and Skärholmen), together accounting for 12.4% of the total population. These differential distributions justified the use of decomposition analysis to quantify the composition effect.

### Association analysis of COVID-19 hospitalization between immigrants by region of birth vs Swedish-born

Table 2 shows the estimates derived from logistic regression models fitted to examine how the association between COVID-19 hospitalization and country/region of birth changes when adjusting for the factors presented in Table 1. First, the unadjusted models show that the odds ratios comparing immigrant groups to the Swedish-born were all statistically significant, for both waves. For example, immigrants from *Middle East* had more than five times higher risk of hospitalization during Wave 1 compared to the Swedish-born (Table 2). Second, after adjusting for all included factors, the odds ratios for immigrant groups all decreased, but almost all remained significantly increased compared to the Swedish-born. In addition, the adjusted models show that older age, male sex (Wave 1), pre-existing medical conditions, essential workers, unemployed or unregistered occupation (Wave 2), high income (Wave 2), low education level, and residence in certain districts (Wave 1) is significantly associated with higher odds of COVID-19 hospitalization.

### Decomposition of factors contributing to COVID-19 hospitalization disparities between immigrants by region of birth and Swedish-born

To quantify the contribution of each factor to the differences in risk between immigrants and Swedish-born, we next performed the decomposition analysis; the results are shown in Figs. 2, 3. Figure 2 shows the risk differences (gray bars on the left panels) and the proportion explained by the composition effect (blue bars on the right panels). For Wave 1, the largest composition effect was observed for *Other European Countries* (48%; percentages regarding the composition effects are given in Supplementary Table S4, S5) followed by *Other Nordic Countries* (44%). Despite having the highest risk differences, *Middle East* and *Africa* exhibit slightly lower explained percentages (18% and 21%, respectively). Only 12% of the difference observed in immigrants from *Rest of the World* is contributed from composition effect. In Wave 2, the risk differences decrease substantially for most groups, except for *Other European Countries*. The composition effects, meanwhile, decrease for all groups. These results show that the differences in the distribution of the seven included factors between Swedish-born and immigrant groups cannot fully explain the disparities in the risk of hospitalization, in particular for immigrants from outside Europe.

Figure 3 shows the contributions of each factor to the risk differences. The seven studied factors (age, sex, pre-existing medical conditions, occupation type, income level, education level, residential district) contribute differently to hospitalization risk differences across immigrant groups. For immigrants from *Other Nordic* and *Other European Countries*, whose composition effects are relatively high, the main contributions come from age (*Other Nordic Countries*: 33% in Wave 1, 31% in Wave 2; *Other European Countries*: 16% in Wave 1, 11 % in Wave 2; percentages given in Supplementary Table S4, S5). This shows that the age structure of these two immigrant groups accounts for a substantial part of the risk difference. For *Other Nordic Countries*, another substantial contribution comes from larger proportion of pre-existing medical conditions (11% in Wave 1, 9% in Wave 2). For *Other European Countries*, additional substantial contributions come from occupation type (11% in Wave 1, 7% in Wave 2) and residential district in Wave 1 (22%). For the other three immigrant groups, the contributions from the seven factors are smaller, but some common patterns emerged. Occupation type contributes significantly to both waves (*Africa*: 5% in Wave 1, 7% in Wave 2; *Middle East*: 3% in Wave 1, 3% in Wave 2; *Rest of the World*: 4% in Wave 1, 4% in Wave 2) and residential district contribute significantly for Wave 1 (*Africa*: 9 %; *Middle East*: 7%; *Rest of the World*: 5%) but not for Wave 2. For all immigrant groups, the magnitude of contribution varies, but some common patterns are observed. Age and occupation type contribute significantly to all the groups for both waves. Residential district contributes positively to Wave 1 for all groups, but the contribution dramatically decreases and becomes non-significant in Wave 2. We also note that some previously studied factors, income and education, have limited or no significant contributions to the composition effect.

Since the composition effects are relatively small for groups with high differences in hospitalization risks, (e.g., *Africa* and *Middle East*), the coefficient effects will be high for these groups. To better understand the contributions to the coefficient effects, we ran logistic regression models for each

**Table 1 | Hospitalization risks (Stockholm, March 1, 2020 to January 31, 2021) and descriptive statistics for the study population (Wave 1) from different country/region of birth. Except population and age, all values are in percentage**

| Characteristics | All Population | Sweden | Africa | Middle East | Other Nordic Countries | Other European Countries | Rest of the World |
|---|---|---|---|---|---|---|---|
| Population | 577,911 | 438,218 | 24,442 | 36,224 | 10,214 | 34,590 | 34,223 |
| Person-time, total amount (average) | 289 096 (0.5002) | 219,362 (0.5006) | 12.184 (0.4985) | 18,049 (0.4983) | 5106 (0.5000) | 17,304 (0.5003) | 17,090 (0.4994) |
| Percentage of Total Population | | 75.8 | 4.2 | 6.3 | 1.8 | 6.0 | 5.9 |
| Hospitalization Incidence in Percentage | | | | | | | |
| Wave 1 | 0.23 | 0.14 | 0.68 | 0.79 | 0.32 | 0.21 | 0.49 |
| Wave 2 | 0.15 | 0.09 | 0.29 | 0.48 | 0.25 | 0.18 | 0.30 |
| Sex | | | | | | | |
| Female | 50.4 | 50.1 | 48.8 | 46.4 | 61.8 | 52.2 | 54.3 |
| Male | 49.6 | 49.9 | 51.3 | 53.6 | 38.2 | 47.8 | 45.7 |
| Age | | | | | | | |
| Mean | 41.0 | 40.2 | 42.4 | 44.2 | 49.1 | 43.0 | 42.3 |
| SD | 12.9 | 13.1 | 11.7 | 12.0 | 12.2 | 11.6 | 11.3 |
| Pre-existing Medical Conditions | | | | | | | |
| No | 90.7 | 91.0 | 88.5 | 87.8 | 85.1 | 92.2 | 90.5 |
| Yes | 9.4 | 9.0 | 11.5 | 12.2 | 14.9 | 7.8 | 9.5 |
| Occupation Type | | | | | | | |
| Essential | 20.2 | 17.1 | 44.9 | 30.7 | 17.9 | 21.0 | 30.1 |
| Non-essential | 57.7 | 63.2 | 24.1 | 34.4 | 62.3 | 50.7 | 42.4 |
| Unemployed or Unregistered | 22.1 | 19.7 | 31.0 | 34.9 | 19.9 | 28.3 | 27.6 |
| Income Level | | | | | | | |
| Low | 33.3 | 30.1 | 48.7 | 47.3 | 27.6 | 40.6 | 44.1 |
| Medium | 33.3 | 32.6 | 39.3 | 35.2 | 33.8 | 33.5 | 36.7 |
| High | 33.3 | 37.4 | 12.0 | 17.5 | 38.7 | 25.9 | 19.3 |
| Education Level | | | | | | | |
| Primary | 10.9 | 8.9 | 27.4 | 23.0 | 7.4 | 8.5 | 15.3 |
| Upper secondary | 31.7 | 31.6 | 42.2 | 34.8 | 28.9 | 26.0 | 30.0 |
| Post secondary | 56.0 | 59.0 | 27.2 | 40.3 | 61.1 | 56.9 | 51.3 |
| Missing | 1.4 | 0.5 | 3.2 | 1.9 | 2.6 | 8.6 | 3.4 |
| District | | | | | | | |
| Bromma | 5.5 | 5.9 | 2.3 | 3.5 | 6.0 | 4.9 | 4.8 |
| Brännkyrka | 5.5 | 5.6 | 4.0 | 5.0 | 5.0 | 5.3 | 6.0 |
| Enskede | 4.6 | 5.1 | 2.4 | 2.2 | 4.5 | 3.2 | 3.8 |
| Essinge | 1.0 | 1.2 | 0.2 | 0.3 | 1.3 | 0.9 | 0.7 |
| Farsta | 6.0 | 5.9 | 6.6 | 5.5 | 5.3 | 6.2 | 8.1 |
| Hägersten | 8.1 | 9.0 | 2.7 | 3.9 | 8.4 | 7.2 | 6.1 |
| Hässelby | 3.8 | 3.1 | 7.1 | 7.4 | 4.0 | 5.4 | 4.8 |
| Högalid | 3.4 | 3.9 | 0.7 | 1.0 | 3.7 | 2.4 | 2.1 |
| Kista | 3.3 | 1.5 | 12.5 | 13.3 | 2.6 | 4.4 | 7.4 |
| Kungsholmen | 2.1 | 2.4 | 0.4 | 0.7 | 2.2 | 1.5 | 1.2 |
| Skarpnäck | 5.2 | 5.5 | 3.8 | 2.6 | 5.7 | 5.0 | 4.9 |
| Skärholmen | 3.6 | 2.0 | 9.0 | 12.9 | 2.1 | 7.1 | 7.3 |
| Spånga | 5.3 | 3.1 | 27.5 | 15 | 3.9 | 6.2 | 6.6 |
| Stockholms Adolf Fredrik | 1.0 | 1.1 | 0.2 | 0.5 | 1.0 | 0.9 | 0.6 |
| Stockholms Engelbrekt | 3.0 | 3.2 | 0.7 | 1.3 | 3.4 | 3.7 | 2.7 |
| Stockholms Gustav Vasa | 1.6 | 1.8 | 0.3 | 0.6 | 1.7 | 1.4 | 0.9 |
| Stockholms Hedvig Eleonora | 1.1 | 1.3 | 0.2 | 0.4 | 1.2 | 0.9 | 0.5 |
| Stockholms Katarina | 3.9 | 4.5 | 1.2 | 1.3 | 4.3 | 2.7 | 2.5 |
| Stockholms Maria Magdalena | 2.1 | 2.4 | 0.5 | 0.6 | 2.4 | 1.5 | 1.4 |
| Stockholms Oscar | 3.9 | 4.5 | 0.5 | 1.2 | 4.0 | 2.9 | 2.1 |

**Table 1 (continued) | Hospitalization risks (Stockholm, March 1, 2020 to January 31, 2021) and descriptive statistics for the study population (Wave 1) from different country/region of birth. Except population and age, all values are in percentage**

| Characteristics | All Population | Sweden | Africa | Middle East | Other Nordic Countries | Other European Countries | Rest of the World |
|---|---|---|---|---|---|---|---|
| Stockholms Sankt Göran | 3.5 | 5.2 | 1.1 | 2.7 | 5.3 | 3.9 | 3.4 |
| Stockholms Sankt Johannes | 1.3 | 1.4 | 0.3 | 0.6 | 1.2 | 1.3 | 0.9 |
| Stockholms Sankt Matteus | 3.5 | 4.0 | 0.5 | 1.2 | 3.6 | 2.8 | 2.1 |
| Stockholms Sofia | 4.2 | 4.7 | 0.7 | 1.4 | 4.9 | 3.3 | 2.7 |
| Stockholms domkyrkodistrikt | 0.5 | 0.6 | 0.1 | 0.3 | 0.6 | 0.6 | 0.4 |
| Vantör | 5.5 | 4.7 | 8.6 | 6.5 | 4.7 | 8.3 | 9.7 |
| Vällingby | 3.8 | 3.2 | 5.7 | 7.6 | 3.6 | 4.4 | 5.1 |
| Västerled | 2.9 | 3.4 | 0.5 | 0.7 | 3.3 | 2.1 | 1.3 |

Descriptive data corresponding to Wave 2 can be found in the Supplementary Information Table S3.

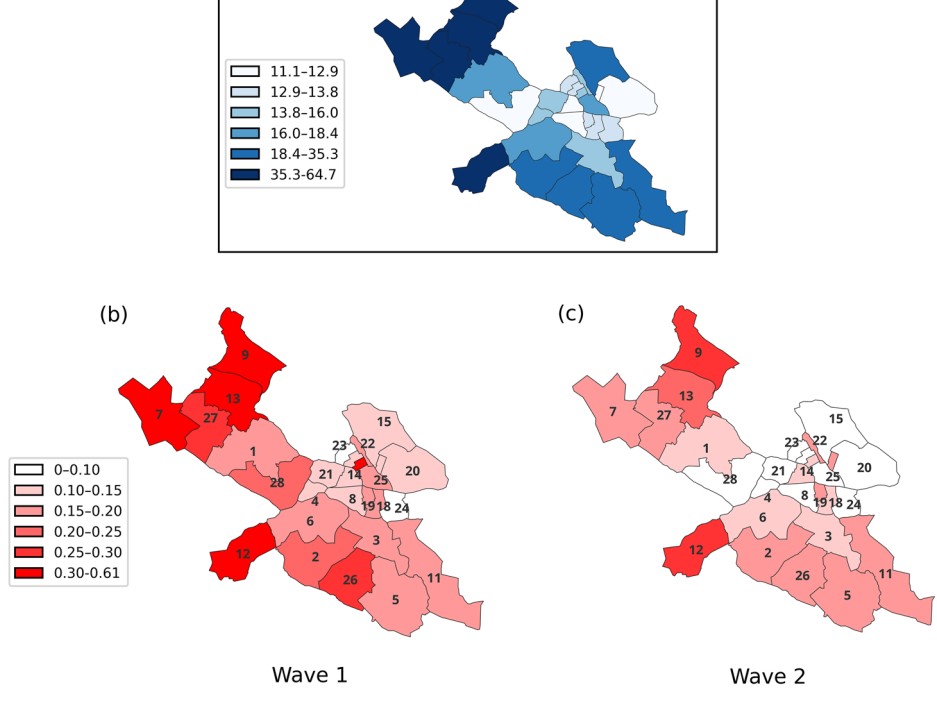

**Fig. 1 | Percentage of immigrants and hospitalization risk by district and wave.** Percentage of immigrants in each district (**a**); hospitalization risk (in %) in each district for (**b**) Wave 1 and (**c**) Wave 2. Districts: 1. Bromma; 2. Brännkyrka; 3. Enskede; 4. Essinge; 5. Farsta; 6. Hägersten; 7. Hässelby; 8. Högalid; 9. Kista; 10. Kungsholmen; 11. Skarpnäck; 12. Skärholmen; 13. Spånga; 14. Stockholms Adolf Fredrik; 15. Stockholms Engelbrekt; 16. Stockholms Gustav Vasa; 17. Stockholms Hedvig Eleonora; 18. Stockholms Katarina; 19. Stockholms Maria Magdalena; 20. Stockholms Oscar; 21. Stockholms Sankt Göran; 22. Stockholms Sankt Johannes; 23. Stockholms Sankt Matteus; 24. Stockholms Sofia; 25. Stockholms domkyrkodistrikt; 26. Vantör; 27. Vällingby; 28. Västerled.

group separately (Fig. S2 in the Supplementary file). The wide range of odds ratios shown in this figure suggests that a given factor may contribute to the hospitalization risk differently for the different groups. The coefficients for several residential district for *Africa* and *Middle East* are higher compared to the Swedish-born population, indicating that living in certain districts carries additional risks for immigrants compared to the Swedish-born. Another interesting thing to note is that the risk of people without education records is remarkably higher among the Swedish-born than among the overall population or immigrant groups. This may suggest that, despite being categorized similarly in terms of educational attainment, Swedish-born individuals without education records differ in important ways from their immigrant counterparts—differences that may contribute to their elevated risk.

## Discussion

Our results show that immigrant groups had much higher risk of hospitalization for COVID-19 compared to the Swedish-born. The main aim of our study was to examine the extent to which the differences in the distribution of factors (group composition) explain these differential risks. The results

showed that group composition contributed to varying degrees and with different patterns in different groups of immigrants. Overall, group composition contributed less to the (large) disparities in hospitalization among the most affected immigrant groups compared to the Swedish-born (21%, 16% for *Africa* and 18%, 11% for *Middle East*, in Wave 1 and Wave 2, respectively). In contrast, it contributed considerably more to the (small) disparities in hospitalization among *Other European* and *Other Nordic Countries* compared to the Swedish-born (48%, 44%, for Wave 1 and 13%, 39% for Wave 2, respectively). Assuming that occupation type and residential district primarily indicate exposure to the virus, and age and pre-existing medical conditions indicate susceptibility, our results suggest a combination of exposure and susceptibility explains differences for *Other European Countries* and non-European immigrants, while susceptibility plays a greater role for Nordic immigrants. In fact, for the latter group, pre-existing medical conditions explain more than forty percent of the hospitalization risk difference.

Our study examines whether the distribution of social determinants explains group differences. The decomposition analysis allowed us to explicitly estimate how the differential distributions of each factor

**Table 2 | Estimates from univariable and multivariable logistic regression models from relevant factors by wave**

| Factors | Wave 1 | | | | Wave 2 | | | |
|---|---|---|---|---|---|---|---|---|
| | Univariable OR (95% CI) | | Multivariable OR (95% CI) | | Univariable OR (95% CI) | | Multivariable OR (95% CI) | |
| **Country/region of Birth** | | | | | | | | |
| Sweden | Reference | | Reference | | Reference | | Reference | |
| Africa | 5.00*** | (4.21, 5.95) | 2.76*** | (2.26, 3.37) | 3.11*** | (2.42, 4.00) | 2.34*** | (1.77, 3.09) |
| Middle East | 5.80*** | (5.04, 6.69) | 3.17*** | (2.69, 3.75) | 5.05*** | (4.23, 6.03) | 3.48*** | (2.83, 4.29) |
| Other Nordic Countries | 2.37*** | (1.67, 3.37) | 1.48* | (1.04, 2.11) | 2.69*** | (1.81, 4.00) | 1.63* | (1.09, 2.43) |
| Other European Countries | 1.57*** | (1.23, 2.00) | 1.23 | (0.96, 1.59) | 1.89*** | (1.45, 2.47) | 1.70*** | (1.29, 2.24) |
| The Rest | 3.61*** | (3.04, 4.29) | 2.76*** | (2.30, 3.32) | 3.12*** | (2.51, 3.87) | 2.73*** | (2.16, 3.43) |
| **Age** | | | | | | | | |
| Age | | | 1.13*** | (1.08, 1.18) | | | 1.13*** | (1.07, 1.19) |
| Age^2 | | | 1.00** | (1.00, 1.00) | | | 1.00* | (1.00, 1.00) |
| **Sex** | | | | | | | | |
| Female | | | Reference | | | | Reference | |
| Male | | | 1.77*** | (1.58, 1.98) | | | 1.50*** | (1.30, 1.73) |
| **Pre-existing Medical Conditions** | | | | | | | | |
| No | | | Reference | | | | Reference | |
| Yes | | | 2.40*** | (2.11, 2.72) | | | 2.31*** | (1.98, 2.70) |
| **Occupation Type** | | | | | | | | |
| Not Frontline | | | Reference | | | | Reference | |
| Frontline | | | 1.52*** | (1.31, 1.75) | | | 1.36** | (1.14, 1.63) |
| Missing | | | 1.17 | (0.99, 1.37) | | | 1.26* | (1.04, 1.54) |
| **Income Level** | | | | | | | | |
| High | | | Reference | | | | Reference | |
| Medium | | | 0.98 | (0.84, 1.14) | | | 0.95 | (0.79, 1.13) |
| Low | | | 0.96 | (0.81, 1.13) | | | 0.81* | (0.65, 1.00) |
| **Education Level** | | | | | | | | |
| Post secondary | | | Reference | | | | Reference | |
| Upper secondary | | | 1.23** | (1.09, 1.40) | | | 1.18* | (1.01, 1.39) |
| Primary | | | 1.39*** | (1.17, 1.64) | | | 1.39** | (1.13, 1.72) |
| Missing | | | 1.50* | (1.00, 2.26) | | | 0.96 | (0.51, 1.82) |
| **District** | | | | | | | | |
| Bromma | | | Reference | | | | Reference | |
| Brännkyrka | | | 1.17 | (0.82, 1.69) | | | 1.07 | (0.72, 1.59) |
| Enskede | | | 1.17 | (0.80, 1.73) | | | 0.92 | (0.59, 1.43) |
| Essinge | | | 1.07 | (0.52, 2.17) | | | 0.27 | (0.06, 1.10) |
| Farsta | | | 0.86 | (0.59, 1.25) | | | 1.11 | (0.76, 1.63) |
| Hägersten | | | 1.22 | (0.86, 1.72) | | | 0.85 | (0.57, 1.27) |
| Hässelby | | | 1.29 | (0.90, 1.85) | | | 0.77 | (0.49, 1.20) |
| Högalid | | | 0.93 | (0.59, 1.48) | | | 0.72 | (0.42, 1.23) |
| Kista | | | 1.32 | (0.93, 1.88) | | | 0.86 | (0.57, 1.32) |
| Kungsholmen | | | 0.71 | (0.39, 1.31) | | | 0.79 | (0.42, 1.47) |
| Skarpnäck | | | 1.07 | (0.73, 1.56) | | | 1.07 | (0.71, 1.60) |
| Skärholmen | | | 1.63** | (1.16, 2.29) | | | 0.98 | (0.66, 1.47) |
| Spånga | | | 1.86*** | (1.35, 2.56) | | | 0.80 | (0.54, 1.19) |
| Stockholms Adolf Fredrik | | | 2.01* | (1.16, 3.50) | | | 0.79 | (0.34, 1.86) |
| Stockholms Engelbrekt | | | 0.87 | (0.52, 1.46) | | | 0.49* | (0.25, 0.97) |
| Stockholms Gustav Vasa | | | 0.80 | (0.42, 1.54) | | | 0.65 | (0.30, 1.38) |
| Stockholms Hedvig Eleonora | | | 1.04 | (0.53, 2.05) | | | 1.25 | (0.65, 2.43) |
| Stockholms Katarina | | | 1.21 | (0.81, 1.81) | | | 0.79 | (0.49, 1.29) |
| Stockholms Maria Magdalena | | | 0.93 | (0.55, 1.57) | | | 1.14 | (0.68, 1.93) |

**Table 2 (continued) | Estimates from univariable and multivariable logistic regression models from relevant factors by wave**

| Factors | Wave 1 | | Wave 2 | |
|---|---|---|---|---|
| | Univariable OR (95% CI) | Multivariable OR (95% CI) | Univariable OR (95% CI) | Multivariable OR (95% CI) |
| Stockholms Oscar | | 0.87 (0.54, 1.39) | | 0.75 (0.44, 1.26) |
| Stockholms Sankt Göran | | 0.80 (0.52, 1.25) | | 0.71 (0.44, 1.17) |
| Stockholms Sankt Johannes | | 1.26 (0.70, 2.28) | | 1.42 (0.78, 2.59) |
| Stockholms Sankt Matteus | | 0.78 (0.47, 1.29) | | 0.83 (0.49, 1.41) |
| Stockholms Sofia | | 0.67 (0.41, 1.08) | | 0.75 (0.46, 1.23) |
| Stockholms domkyrkodistrikt | | 0.88 (0.35, 2.21) | | 0.58 (0.18, 1.88) |
| Vantör | | 1.27 (0.90, 1.80) | | 0.94 (0.63, 1.40) |
| Vällingby | | 1.18 (0.81, 1.72) | | 0.90 (0.59, 1.39) |
| Västerled | | 1.37 (0.89, 2.11) | | 0.65 (0.36, 1.17) |

All tests were two-sided. No adjustment was made for multiple comparisons.

*$p < 0.05$; **$p < 0.01$; ***$p < 0.001$.

contribute to the differences in hospitalization risk across groups. The small overall composition effect observed among immigrants from Africa and Middle East (i.e., from the origins most affected by the pandemic) means that the differential distribution of the included factors did not fully explain the risk differences. Such phenomenon does not only appear in Swedish context, but has been seen in studies conducted in other Nordic countries[19,20]. Therefore, further studies are needed to identify additional factors, such as social contacts, that may account for the differences in risks between immigrant groups and the local-born. Several aspects of social contacts are relevant for disease transmissions, not only the number of interactions but also the type[21–23]. Moreover, given the nonlinear nature of infectious disease outbreaks, even small differences in social contacts may play a major role in disease spread in communities[24–26]. Social contacts may also influence viral load, which has been shown to increase the risk of hospitalization[27]. However, these contacts are hard to measure at the population level, and it is not known if differences in the social contacts between immigrant groups and the Swedish-born can explain the differential disease transmission. This remains an important line of research for future studies.

Although most factors contribute relatively equally across both waves, residential district was a notable exception: it had a positive contribution for all groups in Wave 1 and none in Wave 2. This is an interesting finding, and is likely due to the geographical clustering of multiple factors increasing community transmission early in the pandemic. Indeed, the two most affected immigrant groups during the pandemic had a high proportion of essential workers, were clustered in a few residential districts, and were also more likely to live in multigenerational households[8]. This combination likely increased the likelihood of exposure and facilitated both within-household and community transmission, thereby raising hospitalization risks in these districts. The contributions of residential districts diminish during Wave 2, likely reflecting a more uniform spread of the virus in this wave, something also indicated by the distribution of hospitalization risks shown in Fig. 1.

The relatively small contribution of pre-existing medical conditions to differences in the risk of hospitalization among immigrants from *Africa* and *Middle East* compared to the Swedish-born is noteworthy. To our knowledge, this is the first study that includes also health conditions (HIV, tuberculosis, and sickle cell disease) related to severe COVID-19 disease, and known to be more prevalent among immigrants[28]. The fact that pre-existing medical conditions contribute only a small portion to differences in the risk of hospitalization may be explained by the fact that the health indicator we use is only a partial indicator of pre-existing medical conditions (i.e., only based on specialized out-patient and in-patient care). Overall, this result gives further support to the interpretation that susceptibility was not the main driver of the higher hospitalization risk among immigrants[28].

Regarding exposure, the factors we used do not include social contacts, likely to be of high importance for transmission. As such, the relatively small contribution of the factors indicative of exposure in our study cannot be used to rule out the possibility that differential exposure was a main driver.

Our study uses high-quality, population-based national register data, and, to the best of our knowledge, the first to link COVID-19 hospitalization outcomes with some key socioeconomic factors and pre-existing medical conditions while incorporating both individual and contextual risk factors. By doing so, we provide a comprehensive overview of the key determinants of hospitalization disparities between immigrants and the Swedish-born. Furthermore, while traditional logistic regression analyses can determine whether differences between groups persist after adjusting for covariates, they do not quantify the relative contribution of each factor to the overall differences between groups. In contrast, the decomposition analysis quantifies how much of the observed differences in hospitalization risks between immigrants and Swedes can be attributed to group composition. This quantification indirectly allows us to identify the contribution of each factor or determinant. Importantly, the decomposition analysis allows for examining the heterogeneity across immigrant groups, revealing how some determinants (such as occupation type and residential areas) can have a significant impact on hospitalization risk differences in certain immigrant groups, while others (such as income and education) have relatively little effect across all groups.

Our study also has limitations. First, our categorization of certain variables may have resulted in grouping individuals with different risks into the same category. In terms of occupation, for example, taxi driver and medical doctors are both categorized as essential occupations, but taxi drivers likely had less access to protective measures compared to medical doctors, thereby having a higher risk of being infected. Similarly, individuals without registered occupations, a category more prevalent among immigrants, may also consist of occupations with different risks. The same limitation may also apply to the categorization of education. For example, Swedish-born individuals classified in the "missing" category show remarkably higher risks compared to those in the same category for migrant groups (Fig. S2 in the Supplementary file). The higher percentage of individuals without education records among immigrants may partly reflect the extra steps involved in verifying or transferring foreign educational qualifications to Swedish registers. Therefore, many immigrants under this category may have actually completed some form of formal education. However, regarding Swedish-born individuals without education records, it is possible that a substantial proportion were unable to complete formal education due to various disabilities—disabilities that may also have increased their risk of severe COVID-19 complications. As a result, the exposure risks for the Swedish-born and immigrants under the same category can be largely different. These shortcomings could lead to a biased

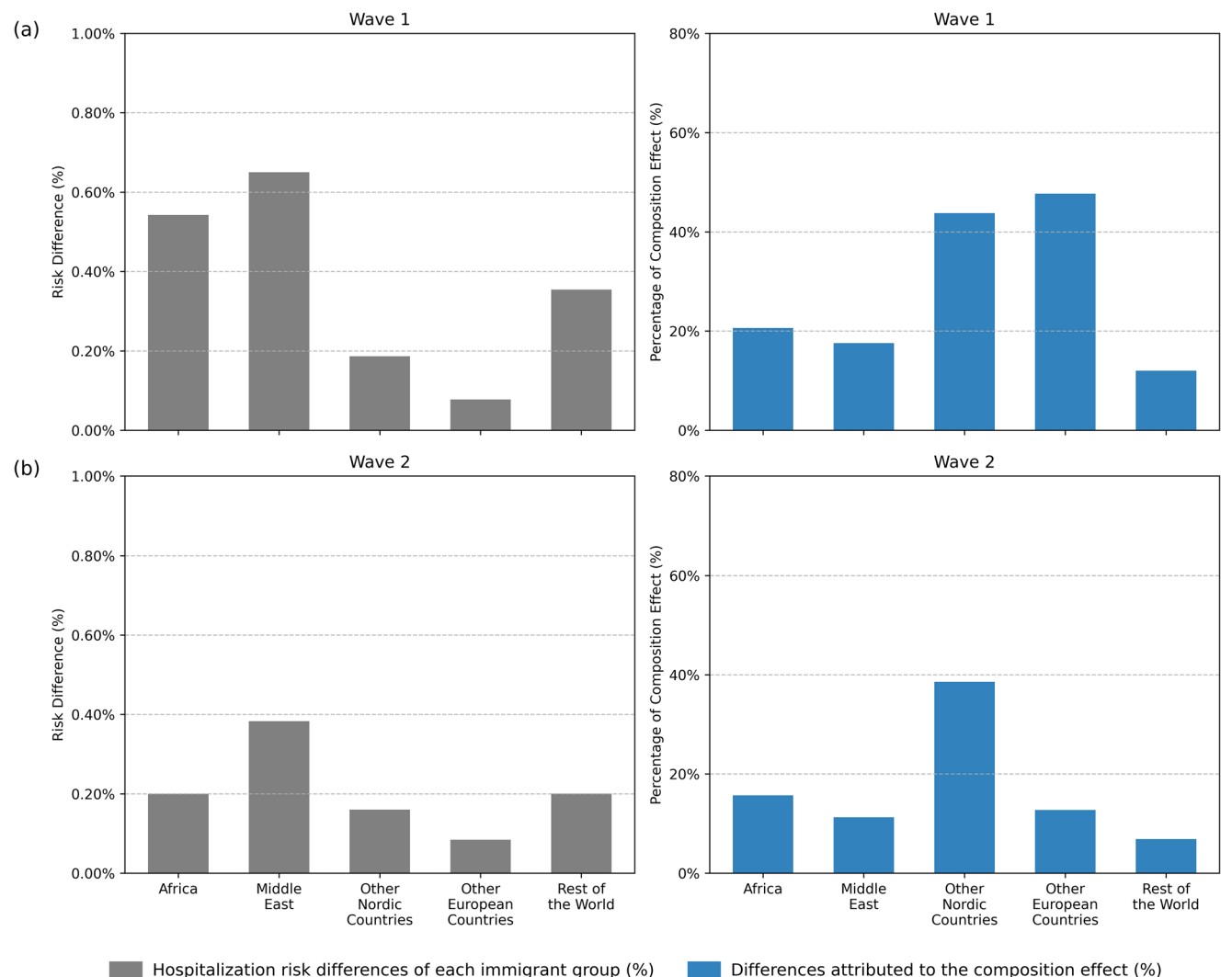

**Fig. 2 | Total and explained share of hospitalization risk differences between Swedish-born individuals and immigrants.** Hospitalization risk differences (percentages) between Swedish-born individuals and immigrants by region of birth, and the percentage of these differences attributed to the composition effect for (**a**) Wave 1 and (**b**) Wave 2. Data are available in Supplementary Information Tables S4, S5.

estimation of the composition effect. Second, the outcome used (hospitalization) reflects a combination of exposure and susceptibility, and given the available data, we cannot quantify the relative importance of the two with precision. Third, a common limitation of migration studies is the under-reporting of emigration to the authorities, which can lead to an artificial health advantage. However, the travel restrictions during the pandemic suggest that this limitation may be less problematic in our study. Finally, although concerns about misclassification in COVID-19 mortality among immigrants have been raised in studies conducted in Sweden[29], we do not expect such misclassification to substantially affect hospitalization data. During the pandemic, individuals in hospitals were more likely to be tested, which reduced the likelihood of misclassification. However, we cannot rule out the possibility that certain barriers to healthcare access some immigrants may have led to an underestimation of hospitalizations. In any case, this limitation is unlikely to affect the overall conclusions of our study.

Although immigrants in other receiving contexts have also shown higher risks of COVID-19 mortality and morbidity, the results of our study should not be generalized to other settings. In fact, further studies should be conducted to determine whether similar patterns are observed in other national contexts and among other immigrant groups. Such as research would further contribute to understanding the contribution of social determinants to COVID-19 disparities by nativity.

Overall, immigrant background has been shown to be an important risk factor for COVID-19 mortality and morbidity. Our study shows that the increased risk can only partly be accounted for by the differential distribution of the observed social determinants often used in public health research.

## Data availability
Individual data cannot be made publicly available due to privacy regulations. According to the General Data Protection Regulation, the Swedish law SFS 2018:218, the Swedish Data Protection Act, the Swedish Ethical Review Act, and the Public Access to Information and Secrecy Act, these types of sensitive data can only be made available for specific purposes, including research, that meets the criteria for access to this type of sensitive and confidential data as determined by a legal review. Individual data are pseudonymized by the Swedish authorities and the key is inaccessible to researchers. Pseudonymized individual data is available from the authors upon request once ethical approval has been obtained, and must be analysed at Stockholm University. Request by access should be sent by email to Sol P Juárez (sol.juarez@su.se). The authors commit to responding to all data-access requests within 30 days of receipt and supplying data (Subject to completion of necessary agreements) within 90 days. Aggregate data used to generate the figures in the main text are provided in Supplementary Data 1,

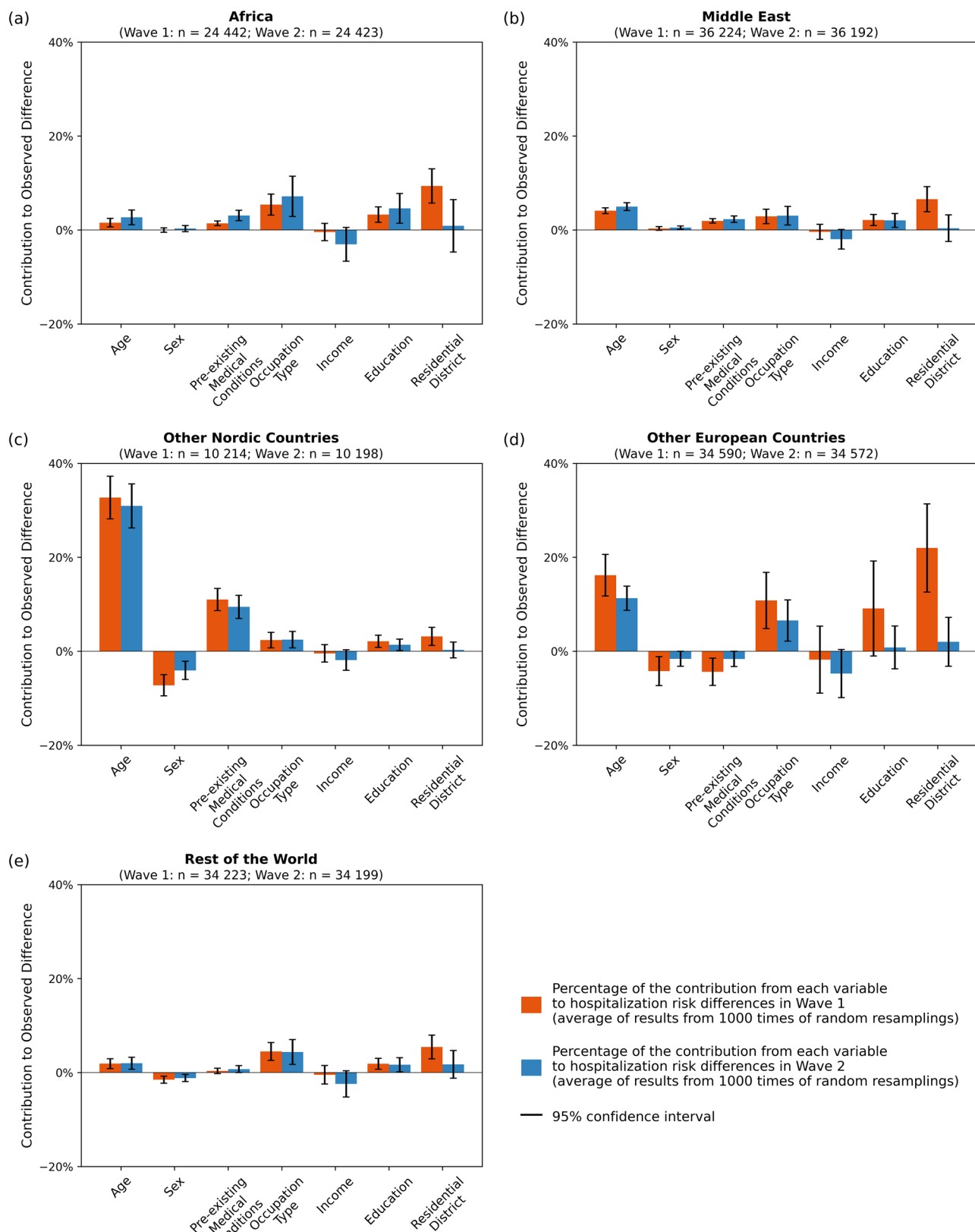

**Fig. 3 | Contributions of factors to hospitalization risk differences.** Contribution of each factor to the hospitalization risk difference (bars reflect estimated attributable percentages with 95% of confidence interval) between Swedish-born individuals (Wave 1: *n* = 438,218; Wave 2: n = 437,953) and immigrants categorized by region of birth: **a** Africa, **b** Middle East, **c** Other Nordic Countries, **d** Other European Countries and (e) the Rest of the World. All data are available in Supplementary Information Tables S4, S5.

and the data used to generate the figures in the Supplementary Information are provided in Supplementary Data 2. In both files, the sheet names correspond to the associated figure numbers.

## Code availability

The code for this study was produced using Python 3.12 and STATA MP 15.1, and is available on Zenodo under a Creative Commons license[30].

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

## Acknowledgements

This study is financed by the Swedish Research Council for Health, Working Life and Welfare (FORTE #2021-00271).

## Author contributions

S.P.J., Y.M., and A.L. conceived the study; Y.M. performed the analyses in consultation with A.L. and S.P.J. Y.M. wrote the original draft, and A.L. and S.J. contributed substantial additional writing, with critical input from SA. Y.M. was responsible for the visualization of the results. All authors had the opportunity to review and revise the manuscript and approve the submitted version.

## Funding

## Competing interest

The authors declare no competing interests.
