## [Transparent Peer Review file · Communications Medicine]

The role of social determinants in COVID-19 hospitalization disparities by migration status in Stockholm, Sweden: A population-based cohort study

Corresponding Author: Dr Sol Juarez

Version 0:

Reviewer comments:

Reviewer #1

(Remarks to the Author)

The subject addressed by the authors is highly relevant for public health researchers in Stockholm, Sweden as well as in any County anywhere such an analysis can be carried out (due to register data availability). In the following I will focus on the methods.

I find the methodological approach chosen in the statistical analysis appropriate and it is based on known methodology described in a couple of articles cited in the references. Nevertheless, some of the steps in the development of the data analysis require more details. The part named Supplementary 1 aims to detail the methodological steps. Yet, there are gaps in the description. For instance, in Equation (7), I am not sure about what is referred to as the set of pooled coefficients, β^* . Is that the set of factor effects estimated in the logistic regression with all individuals in the study sample (both Swedish born and not) as well as regarding a dummy variable for each group of countries? There are a few possibilities of regression model specifications in this context so as to generate the so called pooled coefficients, hence a clearer description (or rather a clearer notation) should be added.

In addition, it is not clear what is the difference between the left and right plots in Figure S1.

Apart from these comments, I think the article is sound and well written.

Reviewer #2

(Remarks to the Author)

Recommendation: Revise

Main Strengths

The paper is clearly written, explores an interesting (and partially understudied) question around how pre-existing social and health inequalities interact with a mortality shock such as the COVID-19 pandemic, uses high-quality data, and adopts a convincing design.

Main Weaknesses

The results of the paper should be better integrated in the existing literature on the topic. It is not clear why the study ends on January 2021, given that we are now in 2025. Figure 2 is not well designed. Potential differences in the coverage of inpatient hospital records by nativity (and other characteristics) should be discussed. Similarly, potential differences in healthcare utilization by nativity (and other characteristics) should be discussed. Could there be a bias if foreign-born individuals are either less likely to be in the inpatient hospital registers or less likely to access the healthcare system (and be hospitalized) relative to Swedish-born individuals? Although it might be outside of the scope of this work, using COVID-19 mortality as a complementary metric might ease both concerns raised above, and potentially provide an interesting test of how outcomes of hospitalization (death or survival) differ by group. The lack of a before-after comparison (before and after the pandemic) makes the interpretation of the results challenging.

Detailed Comments

Although not very large, there is now some literature on differential mortality by nativity during the COVID-19 pandemic which this paper should cite and discuss (Bacong et al. 2024; Horner, Wrigley-Field, and Leider 2022; Paglino and Elo 2024, 2025; Riley et al. 2021). It would make the results better integrated in the existing literature and strengthen the relevance of the study.

It was not clear why the study stopped in January 2021. The fact the vaccines became available would provide an interesting test of how important transmission and exposure patterns were, rather than representing a limitation. From the United States, there is evidence that the initial higher mortality of the foreign-born population reversed during later stages of the pandemic (Paglino and Elo 2025), which highlights the importance of examining temporal variation before drawing conclusions and policy recommendations.

In Figure 2, I strongly recommend having separate panels for the two variables being displayed. Having two y-axes in the same figure is confusing and can mislead readers.

Not my area of expertise, although I am a user of Finnish register data, but I know that there are questions about the completeness and quality of inpatient hospital registers including potential for certain interactions being missed and incorrect coding of ICD codes. This should be discussed, at least to say that based on previous studies these issues are of a small enough magnitude that they can be safely ignored. When populations with different sociodemographic characteristics are compared, it is important to think about how their differential access to healthcare can affect the results.

This might be out of scope, but an additional analysis with mortality as an outcome would probably be more robust to differential coverage, and, interestingly, would also allow for a before-after comparison which is not possible when the outcome is COVID-19 hospitalization. What I mean is that right now, it is hard to say whether the differences we are seeing represent a worse impact of COVID-19 on foreign-born populations, because we do not know what differences to expect. With mortality, we typically now that immigrant populations have lower mortality than native-born populations, which makes findings of a foreign-born disadvantage during the pandemic more surprising. Another route to get at the same issue would be to include other types of hospitalization (e.g. for myocardial infarction, for pneumonia) which could be compared across groups both before and after the COVID-19 pandemic.

References

- Bacong, A. M., R. Chu, A. Le, V. Bui, N. E. Wang, and L. P. Palaniappan. 2024. "Increased COVID-19 Mortality among Immigrants Compared with US-Born Individuals: A Cross-Sectional Analysis of 2020 Mortality Data." *Public Health* 231:173–78. doi:10.1016/j.puhe.2024.03.016.
- Horner, Kimberly M., Elizabeth Wrigley-Field, and Jonathon P. Leider. 2022. "A First Look: Disparities in COVID-19 Mortality Among US-Born and Foreign-Born Minnesota Residents." *Population Research and Policy Review* 41(2):465–78. doi:10.1007/s11113-021-09668-1.
- Paglino, Eugenio, and Irma T. Elo. 2024. "Immigrant Mortality Advantage in the United States during the First Year of the COVID-19 Pandemic." *Demographic Research* 50(7):185–204. doi:10.4054/DemRes.2024.50.7.
- Paglino, Eugenio, and Irma T. Elo. 2025. "US-Born and Foreign-Born Life Expectancy by Race and Hispanic Origin before and during the COVID-19 Pandemic in the United States." *Social Science & Medicine* 380:118191. doi:10.1016/j.socscimed.2025.118191.
- Riley, Alicia R., Yea-Hung Chen, Ellicott C. Matthey, M. Maria Glymour, Jacqueline M. Torres, Alicia Fernandez, and Kirsten Bibbins-Domingo. 2021. "Excess Mortality among Latino People in California during the COVID-19 Pandemic." *SSM - Population Health* 15:100860. doi:10.1016/j.ssmph.2021.100860.

Reviewer #3

(Remarks to the Author)

COMMSMED-25-1092-T: The role of social determinants in COVID-19 hospitalization disparities by migration status in Stockholm, Sweden

The paper uses rich registry data from Stockholm to investigate how the correlations between various medical and socio-economic variables and covid-19-related hospitalization differs for Swedish-born and foreign born-residents. They use a version of the Blinder-Oaxaca-decomposition to try to shed light on how much of the correlations that stem from differences in the composition of the groups, and they find that the higher risk of hospitalization among immigrant groups from low- and middle-income countries can only to a small extent be accounted for by differential distribution of the included socioeconomic variables.

I agree with the authors' statement at end of their abstract that, "To be prepared for future pandemics, we need a better understanding of the mechanisms that increase the risk of severe consequences among immigrants". However, I fail to see how their mechanical decomposition contributes much to documenting such mechanisms. In my experience, there is rarely

much empirical insight to be elicited (compared with just doing regressions with and without the covariates) by applying Blinder-Oaxaca-decompositions on such correlations, and this is also my impression after having read this paper.

The authors' motivate their study by stating that, "prior studies have not quantified the contribution of differential distributions of health and social determinants to the increased risk" of hospitalization with covid-19 among immigrants compared with native-born. For example, Labberton et al (Sc. Journal of Public Health, 2022) (<https://journals.sagepub.com/doi/10.1177/14034948221075029>) also compare the risk of hospitalization with covid-19 across immigrant groups and native-born, in models with and without socio-economic covariates (and they, too, seem to find that including the socioeconomic variables affects the results little).

The authors introduce the concepts of "exposure" and "susceptibility", and make very bald assumptions that the prior can be captured by very broad categories of occupation/area of residency and the latter by age and a few indicators of pre-existing medical conditions. I do not see how introducing these concepts bring any insight (in addition to just comment on the estimates for the variables of occupation etc).

I found the authors' definitions of Regional groups (Table S1) hard to follow, like for example "Africa" including "Somalia, Africa,..."

Version 1:

Reviewer comments:

Reviewer #2

(Remarks to the Author)

I thank the authors for their efforts in addressing my comments. I am satisfied with their answers and I feel the revised paper is ready for publication.

Reviewer #3

(Remarks to the Author)

Response to Reviewers' Comments

Dear reviewers,

Thank you very much for your valuable comments. We have revised our manuscript accordingly and provided our point-by-point responses below.

We would like to inform you that during the review process, we identified a mistake in the definition of our outcome. Specifically, we had mistakenly included both in-patient and out-patient data together. Although this does not substantially affect the results, it does impact the number of observations. As a result, you may notice some discrepancies compared to the previous version of the manuscript. We apologize for the inconvenience.

Sincerely,

The authors

In the following sections, reviewers' comments are in italics, and the responses are in regular text.

Responses to reviewer #1:

The subject addressed by the authors is highly relevant for public health researchers in Stockholm, Sweden as well as in any County anywhere such an analysis can be carried out (due to register data availability). In the following I will focus on the methods.

We thank the reviewer for the positive comments!

I find the methodological approach chosen in the statistical analysis appropriate and it is based on known methodology described in a couple of articles cited in the references. Nevertheless, some of the steps in the development of the data analysis require more details.

- 1. The part named Supplementary 1 aims to detail the methodological steps. Yet, there are gaps in the description. For instance, in Equation (7), I am not sure about what is referred to as the set of pooled coefficients, β^* . Is that the set of factor effects estimated in the logistic regression with all individuals in the study sample (both Swedish born and not) as well as regarding a dummy variable for each group of countries? There are a few possibilities of regression model specifications in this context so as to generate the so-called pooled coefficients, hence a clearer description (or rather a clearer notation) should be added.*

Response: Thank you for pointing out this issue. We actually mentioned the definition of pooled coefficient in the section before β^* appears. To be clearer, we have now added " β^* contains pooled coefficients (estimated from logistic regressions including all study populations, both Swedish-born and immigrants, with dummy variables for regional groups)" immediately after β^* is introduced. This addition is highlighted on page 3 of the Supplementary file.

- 2. In addition, it is not clear what is the difference between the left and right plots in Figure S1.*

Response: Thank you for noting this. We apologize that the figure was not appropriately labeled. The left panel shows the results corresponding to Wave 1 and the right panel is for the results for Wave 2. We have now clarified this in the new version.

The caption of this figure was changed to: "Odds ratios from logistic regression for each country-of-origin group and entire population for Wave 1 (left panel), and Wave 2 (right panel). Odds ratios with a p-value less than 0.05 are marked with filled circle, those with a p-value greater than 0.05 are marked with empty circles". This change is highlighted on page 7 of the Supplementary file.

Apart from these comments, I think the article is sound and well written.

We thank the reviewer for the positive comments.

Responses to Reviewer #2:

Recommendation: Revise

Main Strengths

The paper is clearly written, explores an interesting (and partially understudied) question around how pre-existing social and health inequalities interact with a mortality shock such as the COVID-19 pandemic, uses high-quality data, and adopts convincing design.

We thank the reviewer for the positive assessment.

Main Weaknesses (Here the main weaknesses have many overlapping with the later detailed comments, so we shortened some answers for some and gave more details in later comments)

1. *The results of the paper should be better integrated in the existing literature on the topic. It is not clear why the study ends on January 2021, given that we are now in 2025.*

Response: Thank you for your comment. In the revised version we have done our best to include more studies on the topic from outside Sweden in the introduction and discussion.

Regarding the study's end time, we certainly would like to have data up to 2025, but unfortunately, we only have data until 2021. This still gives the possibility of investigating the third wave; however, this wave coincided with the vaccine rollout and therefore we did not include data from this wave in study (please see also comment #6).

2. *Figure 2 is not well designed.*

Response: We apologize if figure 2 was not very clear. In the revised version, we have changed it to make it easier to read, by replacing the double y-axis to single y-axes. Please see also our response to your detailed comment #7.

3. *Potential differences in the coverage of inpatient hospital records by nativity (and other characteristics) should be discussed. Similarly, potential differences in healthcare utilization by nativity (and other characteristics) should be discussed. Could there be a bias if foreign-born individuals are either less likely to be in the inpatient hospital registers or less likely to access the healthcare system (and be hospitalized) relative to Swedish-born individuals?*

Response: Thank you for raising this important issue. First, we would like to point out that any differences in coverage caused by nativity would lead to a downward bias of our results. This means that we could be underestimating the differences between immigrant groups and Swedish-born. However, our population consists of people resident in Sweden for which the coverage of the hospital records is believed to be almost 100% (Ludvigsson et al., 2011). Moreover, although some degree of misclassification may exist especially in the beginning of the pandemic, it is expected to be less significant than for other outcomes (such as mortality) since individuals admitted to hospitals were more likely to be tested. We therefore believe that it is unlikely that differences in coverage or misclassification will affect our results.

Regarding differences in access to healthcare, please see our response to your detailed comment #8.

4. *Although it might be outside of the scope of this work, using COVID-19 mortality as a complementary metric might ease both concerns raised above, and potentially provide an interesting test of how outcomes of*

hospitalization (death or survival) differ by group. The lack of a before-after comparison (before and after the pandemic) makes the interpretation of the results challenging.

Response: We agree that COVID-19 mortality is an important outcome. However, for the working age population that is our focus, mortality is too low to allow for decomposition analysis. In any case, several previous studies have already shown great COVID-19 mortality differences between immigrants and Swedish natives, and we cite those reference in the introduction (see references #5, 7 and 8 in the revised manuscript). Please see our response to your detailed comment #9.

Detailed Comments

- 5. Although not very large, there is now some literature on differential mortality by nativity during the COVID-19 pandemic which this paper should cite and discuss (Bacong et al. 2024; Horner, Wrigley-Field, and Leider 2022; Paglino and Elo 2024, 2025; Riley et al. 2021). It would make the results better integrated in the existing literature and strengthen the relevance of the study.*

Response: Thank you for your comment and the references. We have added them in introduction section:

Lines 21-22: “Immigrants (i.e., foreign-born) in many countries were disproportionately affected by the COVID-19 pandemic ¹⁻⁴.”

Lines 24-26: “The greater impact of the pandemic on immigrants, both in Sweden and abroad, was in stark contrast to their often lower mortality compared to native populations prior to the pandemic ⁹⁻¹².”

Please note that due to space constrains we have not elaborated on the findings, but we hope the inclusion of references helps situate the study in the existing literature.

- 6. It was not clear why the study stopped in January 2021. The fact the vaccines became available would provide an interesting test of how important transmission and exposure patterns were, rather than representing a limitation. From the United States, there is evidence that the initial higher mortality of the foreign-born population reversed during later stages of the pandemic (Paglino and Elo 2025), which highlights the important of examining temporal variation before drawing conclusions and policy recommendations.*

Response: We thank the reviewer for raising this important point. Our original focus on the first two waves (up to January 2021) was motivated by the fact that this period preceded widespread vaccination in Sweden. By restricting the analysis to the pre-vaccination period, we aimed to provide clearer insight into the underlying social and health determinants of COVID-19 hospitalization disparities, without the effects of vaccine uptake, which likely open new pathways through which differences in hospitalization could arise. We agree that these pathways are important to examine but believe that this is beyond the scope of this work.

- 7. In Figure 2, I strongly recommend having separate panels for the two variables being displayed. Having two y-axes in the same figure is confusing and can mislead readers.*

Response: Thanks to give this suggestion. Figure 2 has been changed on the page 21 of the manuscript by separating panels as in the figure below:

Figure 1 (inserted as Figure 2 in the revised manuscript). Hospitalization risk differences of each immigrant group (gray color, corresponding to left y-axis), and percentage of the differences contributed from composition effect (blue color, corresponding to the right y-axis), for (a) Wave 1 and (b) Wave 2. All data can be found in Supplementary Information Table S4 and Table S5 in the Supplementary Information.

8. *Not my area of expertise, although I am a user of Finnish register data, but I know that there are questions about the completeness and quality of inpatient hospital registers including potential for certain interactions being missed and incorrect coding of ICD codes. This should be discussed, at least to say that based on previous studies these issues are of a small enough magnitude that they can be safely ignored. When populations with different sociodemographic characteristics are compared, it is important to think about how their differential access to healthcare can affect the results.*

Response: Thank you for pointing out this important concern. We have discussed the completeness and quality of inpatient hospital registers in comment #3. We completely agree that barriers to access are important when discussing health outcomes among immigrants and it has been overlooked in our previous version. Although Sweden offers universal healthcare access to all individuals registered in the country (our study population), this does not mean that migrants do not face other barriers (such as language difficulties and discrimination) that could directly impact health inequalities, which would lead to an underestimation of the overall differences between immigrants and native Swedes, however this limitation is unlikely to change the result of our study. We have now elaborated on this issue in the discussion section as follows:

Lines 320-326: “Although concerns about misclassification in COVID-19 mortality among immigrants have been raised in studies conducted in Sweden,²⁹ we do not expect such misclassification to substantially affect hospitalization data. During the pandemic, individuals in hospitals were more likely to be tested, which reduced the likelihood of misclassification. However, we cannot rule out the possibility that certain barriers to healthcare access some immigrants may have led to an underestimation of hospitalizations. In any case, this limitation is unlikely to affect the overall conclusions of our study.”

9. *This might be out of scope, but an additional analysis with mortality as an outcome would probably be more robust to differential coverage, and, interestingly, would also allow for a before-after comparison which is not possible when the outcome is COVID-19 hospitalization.*

Response: Thank you for your comment. As mentioned in comment #4, we did not use mortality since the number of events are too low to apply the decomposition analysis (see Table 1 below). Note that logistic regression, as the first step of decomposition analysis, requires about 10 events per variable to give a robust fit (Peduzzi et al., 1996).

Table 1. COVID-19 mortality of Wave 1 and Wave 2 for each country/region group:

Mortality (% (number of death))	Sweden	Africa	Middle East	Other Nordic Countries	Other European Countries	Rest of the World
Wave 1	0.01 (44)	0.03 (8)	0.03 (12)	0.05 (5)	0.01 (2)	0.02 (6)
Wave 2	0.002 (8)	0.004 (1)	0.006 (2)	0.010 (1)	0 (0)	0.006 (2)

Furthermore, several studies have already examined COVID-19 mortality between immigrants and Swedish-born individuals using data from both Stockholm and Sweden (see Drefahl et al., 2020; Juárez et al., 2024; Rostila et al., 2021). Some of these studies also provide comparisons for “all-cause mortality” before the pandemic and “all other causes” during the pandemic. The findings indicate the pandemic reversed the mortality advantage previously observed in most immigrant groups. Moreover, the patterns in “all other causes” during the pandemic suggest that serious problems of misclassification issues are unlikely.

In revised version, we have briefly mentioned the changes in mortality before and after the pandemic, as follows:

Lines 22-26: “In Sweden, immigrants, particularly from low- and middle-income countries, such as Iran, Iraq, and Somalia, had significantly higher mortality rates and more severe complications during the pandemic compared to the Swedish-born⁵⁻⁸. The greater impact of the pandemic on immigrants, both in Sweden and abroad, was in stark contrast to their lower mortality compared to native populations prior to the pandemic⁹⁻¹².”

10. *What I mean is that right now, it is hard to say whether the differences we are seeing represent a worse impact of COVID-19 on foreign-born populations, because we do not know what differences to expect. With mortality, we typically now that immigrant populations have lower mortality than native-born populations, which makes findings of a foreign-born disadvantage during the pandemic more surprising. Another route to get at the same issue would be to include other types of hospitalization (e.g. for myocardial infarction, for pneumonia) which could be compared across groups both and after the COVID-19 pandemic.*

Response: Thank you for your comment. We agree with the reviewer that we have not explicitly compared our results with those from before the pandemic. However, this issue has been partially addressed in previous studies, which showed lower risks of hospitalization among immigrants before the pandemic for most of the conditions associated with severe COVID-19 disease (including chronic respiratory diseases, ICD J40-J47) compared to native Swedes (Juárez et al., 2023). In the revised manuscript, we have added the text, and a supplementary figure of hospitalization risk for infectious and parasitic diseases and for respiratory system diagnoses before and during the pandemic to address this matter, as follows:

Lines 156-159: “These large differences in hospitalization are much greater than what has been observed for infectious and parasitic diseases or for respiratory system diagnoses (excluding COVID-19) before and during the pandemic (Supplementary Fig. S1).”

In Figure 4 below (inserted in Supplementary file as Figure S1), we have compared hospitalization risk for infectious & parasitic diagnosis (ICD -10: A00-B99) as well as respiratory system (ICD-10: J00-J99) for several year before the pandemic and during the pandemic. As it shows below, although before and during the pandemic, immigrants from certain regions (such as immigrants from other Nordic countries) have slightly higher hospitalization risk regarding infectious & parasitic diagnosis and respiratory system, the disparity is not as big as for COVID-19, no matter for which immigrant group.

Hospitalization before the pandemic:

Hospitalization during the pandemic:

Hospitalization from COVID during pandemic:

Figure 4 (inserted in Supplementary file as Figure S1). Hospitalization risk for infectious & parasitic diagnosis (ICD -10: A00-B99) and respiratory system diagnosis (ICD-10: J00-J99) for each year before the pandemic (first row) and during the pandemic (second row) compared with COVID-19 hospitalization risk (third row).

Response to Reviewer #3:

The paper uses rich registry data from Stockholm to investigate how the correlations between various medical and socio-economic variables and covid-19-related hospitalization differs for Swedish-born and foreign born-residents. They use a version of the Blinder-Oaxaca-decomposition to try to shed light on how much of the correlations that stem from differences in the composition of the groups, and they find that the higher risk of hospitalization among immigrant groups from low- and middle-income countries can only to a small extent be accounted for by differential distribution of the included socioeconomic variables.

1. *I agree with the authors' statement at end of their abstract that, "To be prepared for future pandemics, we need a better understanding of the mechanisms that increase the risk of severe consequences among immigrants". However, I fail to see how their mechanical decomposition contributes much to documenting such mechanisms. In my experience, there is rarely much empirical insight to be elicited (compared with just doing regressions with and without the covariates) by applying Blinder-Oaxaca-decompositions on such correlations, and this is also my impression after having read this paper.*

Response: Thank you for this comment. Our motivation for using the decomposition was to better interpret the results of logistic regressions of hospitalizations on a number of explanatory variables. Since hospitalization is a binary outcome, the natural choice of regression model is logistic regression. It is our belief that comparing odds ratios from different regression models is associated with difficulties of interpretation (Mood, 2010). In particular, as we were interested in quantifying the individual contribution of several factors to the hospitalization risk difference, we turned to decomposition analysis that allows us to do precisely that. Perhaps there are other ways to achieve the same goal, but since we use logistic regression, adding and subtracting covariates is not one of them. Consequently, we do not share the reviewer's negative view on the empirical insights that can be gained from a decomposition analysis. Furthermore, we do think that our decomposition results provide information on the mechanisms that increases the risk of severe consequences of COVID-19 in immigrants. It is true that none of the variables that we have investigated alone account for the risk differences, and taken together the variables explain not more than 48% of the differences. We believe this is an important result in itself (it could have turned out to be different), and we disagree with the reviewer that the results don't contribute to our understanding of the mechanisms. Still, we have changed the concluding sentence of the manuscript to: "To be more prepared for future pandemics, we need to further identify the relevant set of social determinants that increase the risk of severe consequences among immigrants", no longer referring to mechanisms.

In the revised version of our manuscript, we also tried to better clarify our motivation to use decomposition analysis:

Lines 124-125: "To quantify the individual contribution of the factors to the hospitalization risk difference, we applied a decomposition analysis."

Lines 290-299: "Furthermore, while traditional logistic regression analyses can determine whether differences between groups persist after adjusting for covariates, they do not quantify the relative contribution of each factor to the overall differences between groups. In contrast, the decomposition analysis quantifies how much of the observed differences in hospitalization risks between immigrants and Swedes can be attributed to group composition. This quantification indirectly allows us to identify the contribution of each factor or determinant. Importantly, the decomposition analysis allows for examining the heterogeneity across immigrant groups, revealing how some determinants (such as occupation type and residential areas) can have a significant impact on hospitalization risk differences in certain immigrant groups, while others (such as income and education) have relatively little effect across all groups"

2. *The authors' motivate their study by stating that, "prior studies have not quantified the contribution of differential distributions of health and social determinants to the increased risk" of hospitalization with covid-19 among immigrants compared with native-born. For example, Labberton et al (Sc. Journal of Public Health, 2022) (<https://journals.sagepub.com/doi/10.1177/14034948221075029>) also compare the risk of hospitalization with covid-19 across immigrant groups and native-born, in models with and without socio-economic covariates (and they, too, seem to find that including the socioeconomic variables affects the results little).*

Response: Thank you for highlighting the study by Labberton et al. (2022), we should have referred to this relevant work in the manuscript. This study, which based on Norway, did capture similar results that adjustment by normal socioeconomic factors explained only 3.8% of the excess hospitalization risk among foreign-born individuals compared to Norwegian-born with native-born parents. For Norwegians with foreign-born parents, the socioeconomic and medical adjustments accounted for 46.2% of the excess hospitalization risk. But our use of decomposition analysis enhances this insight by not only showing that

disparities persist but also quantifying precisely how much each covariate contributes to explaining the risk gap versus how much remains unexplained.

Now we have cited this study in the revised manuscript:

Lines 250-254: “The small overall composition effect observed among immigrants from Africa and Middle East (i.e., from the origins most affected by the pandemic) means that the differential distribution of the included factors did not fully explain the risk differences. Such phenomenon does not only appear in Swedish context, but has been seen in studies conducted in other Nordic countries ^{19,20}.”

3. *The authors introduce the concepts of "exposure" and "susceptibility", and make very bald assumptions that the prior can be captured by very broad categories of occupation/area of residency and the latter by age and a few indicators of pre-existing medical conditions. I do not see how introducing these concepts bring any insight (in addition to just comment on the estimates for the variables of occupation etc).*

Response: We thank the reviewer for this thoughtful comment. We are sorry if our introduction gave the impression that a main focus of our work was to capture the contribution of exposure and susceptibility to the risk differences with these set of variables. We have now completely re-written the second paragraph of the introduction to better align with our aims.

Just to clarify, our intention was not to claim that these variables comprehensively capture the complexity of exposure or susceptibility but rather that their potential effects can be conceptualized (and has been conceptualized) in these terms.

Please check our revised second paragraph of the introduction:

Lines 27-35: “Differences in COVID-19 outcomes between immigrants and the Swedish-born have been reported for both working- and retirement-age populations, with higher risk ratios among the working-age population ⁸. These differences remained after adjusting for health and socio-demographic variables – factors likely reflecting a mixture of differential exposure and susceptibility ^{5,8,10}. However, these studies did not quantify the relative contribution of each factor, nor did examine in detail the possibility that the same set of health and social factors may impact the risk differently for different immigrant groups. In other words, previous studies have not completely acknowledged the heterogeneity of the immigrant groups in Sweden, as they implicitly assume that all socio-demographic variables have the same “effects” across all origins.”

4. *I found the authors' definitions of Regional groups (Table S1) hard to follow, like for example "Africa" including "Somalia, Africa,.."*

Response: We thank the reviewer for noting the potential confusion regarding the regional group definitions in Supplementary Table S1. The groupings follow directly from how country/region of birth we obtained from Statistics Sweden. For some individuals, country of birth is registered at the national level (e.g., Somalia), while for others only a broader regional or continental code is available (e.g., “Africa” or “Middle East”), due to data protection rules and statistical disclosure control. To maintain consistency and comparability, we used the register-defined categories without modification.

We have clarified this in the Supplementary Information under the Table S1 to avoid misunderstanding: “Note: Regional groups reflect the categories we have obtained from Statistics Sweden. For some individuals, country of birth is recorded at the national level (e.g., Somalia), whereas for others only a broader regional or continental code is available (e.g., “Africa” or “Middle East”). These broader categories arise due to statistical secrecy and disclosure control procedures. Our groupings therefore directly follow the register definitions without alteration”.

References:

- Drefahl, S., Wallace, M., Mussino, E., Aradhya, S., Kolk, M., Brandén, M., Malmberg, B., Andersson, G., 2020. A population-based cohort study of socio-demographic risk factors for COVID-19 deaths in Sweden. *Nat. Commun.* 11, 5097. <https://doi.org/10.1038/s41467-020-18926-3>
- Juárez, S.P., Cederström, A., Aradhya, S., Rostila, M., 2023. Differences in hospitalizations associated with severe COVID-19 disease among foreign- and Swedish-born. *Eur. J. Public Health* 33, 522–527. <https://doi.org/10.1093/eurpub/ckad009>
- Juárez, S.P., Debiase, E., Wallace, M., Drefahl, S., Mussino, E., Cederström, A., Rostila, M., Aradhya, S., 2024. COVID-19 mortality among immigrants by duration of residence in Sweden: a population-based cohort study. *Scand. J. Public Health* 52, 370–378. <https://doi.org/10.1177/14034948241244560>
- Ludvigsson, J.F., Andersson, E., Ekbom, A., Feychting, M., Kim, J.-L., Reuterwall, C., Heurgren, M., Olausson, P.O., 2011. External review and validation of the Swedish national inpatient register. *BMC Public Health* 11, 450. <https://doi.org/10.1186/1471-2458-11-450>
- Mood, C., 2010. Logistic Regression : Why We Cannot Do What We Think We Can Do, and What We Can Do About It. *Eur. Sociol. Rev.* 26, 67–82.
- Mussino, E., Drefahl, S., Wallace, M., Billingsley, S., Aradhya, S., Andersson, G., 2024. Lives saved, lives lost, and under-reported COVID-19 deaths: Excess and non-excess mortality in relation to cause-specific mortality during the first year of the COVID-19 pandemic in Sweden. *Demogr. Res.* 50, 1–40. <https://doi.org/10.4054/DemRes.2024.50.1>
- Peduzzi, P., Concato, J., Kemper, E., Holford, T.R., Feinstein, A.R., 1996. A simulation study of the number of events per variable in logistic regression analysis. *J. Clin. Epidemiol.* 49, 1373–1379. [https://doi.org/10.1016/s0895-4356\(96\)00236-3](https://doi.org/10.1016/s0895-4356(96)00236-3)
- Rostila, M., Cederström, A., Wallace, M., Brandén, M., Malmberg, B., Andersson, G., 2021. Disparities in Coronavirus Disease 2019 Mortality by Country of Birth in Stockholm, Sweden: A Total-Population–Based Cohort Study. *Am. J. Epidemiol.* 190, 1510–1518. <https://doi.org/10.1093/aje/kwab057>